# Modulation of Dopamine Receptors on Osteoblasts as a Possible Therapeutic Strategy for Inducing Bone Formation in Arthritis

**DOI:** 10.3390/cells11101609

**Published:** 2022-05-11

**Authors:** Elena Schwendich, Laura Salinas Tejedor, Gernot Schmitz, Markus Rickert, Jürgen Steinmeyer, Stefan Rehart, Styliani Tsiami, Jürgen Braun, Xenofon Baraliakos, Jörg Reinders, Elena Neumann, Ulf Müller-Ladner, Silvia Capellino

**Affiliations:** 1Project Group Neuroimmunology, Department of Immunology, IfADo—Leibniz Research Center for Working Environment and Human Factors, 44139 Dortmund, Germany; schwendich@ifado.de (E.S.); salinastejedor.laura@gmail.com (L.S.T.); gernot.schmitz@tu-dortmund.de (G.S.); 2Department of Orthopaedics and Orthopaedic Surgery, University Hospital Giessen and Marburg (UKGM), 35392 Giessen, Germany; markus.rickert@ortho.med.uni-giessen.de; 3Laboratory for Experimental Orthopaedics, Department of Orthopaedics and Orthopaedic Surgery, Justus Liebig University Giessen, 35392 Giessen, Germany; juergen.steinmeyer@ortho.med.uni-giessen.de; 4Clinic for Orthopedics and Trauma Surgery, Agaplesion Markus Teaching Hospital of Johann Wolfgang Goethe-University, 60431 Frankfurt am Main, Germany; stefan.rehart@fdk.info; 5Rheumazentrum Ruhrgebiet Herne, Ruhr University Bochum, Claudiusstr. 45, 44649 Herne, Germany; styliani.tsiami@elisabethgruppe.de (S.T.); juergen.braun@elisabethgruppe.de (J.B.); xenofon.baraliakos@elisabethgruppe.de (X.B.); 6Analytical Chemistry, Department of Toxicology, IfADo—Leibniz Research Center for Working Environment and Human Factors, 44139 Dortmund, Germany; reinders@ifado.de; 7Kerckhoff-Klinik GmbH, Rheumatology and Clinical Immunology, Justus Liebig University Giessen, 61231 Bad Nauheim, Germany; e.neumann@kerckhoff-klinik.de (E.N.); u.mueller-ladner@kerckhoff-klinik.de (U.M.-L.)

**Keywords:** rheumatoid arthritis, dopamine, osteoblast, osteoclasts, mineralization

## Abstract

Rheumatoid arthritis (RA) is associated with systemic osteoporosis, which leads to severe disability and low quality of life. Current therapies target osteoclasts to reduce bone degradation, but more treatment options would be required to promote bone protection by acting directly on osteoblasts (OB). Recently, the local production of dopamine in inflamed joints of RA has been observed. Thus, in this project, we aimed to determine the implication of the neurotransmitter dopamine in the bone formation process in RA. Dopamine receptors (DR) in the human bone tissue of RA or osteoarthritis (OA) patients were examined by immunohistochemistry. DR in isolated human osteoblasts (OB) was analyzed by flow cytometry, and dopamine content was evaluated by ELISA. Osteoclasts (OC) were differentiated from the PBMCs of healthy controls (HC) and RA patients. Isolated cells were treated with specific dopamine agonists. The effect of dopamine on mineralization was evaluated by Alizarin red staining. Cytokine release in supernatants was measured by ELISA. Osteoclastogenesis was evaluated with TRAP staining. OC markers were analyzed via real-time PCR and bone resorption via staining of resorption pits with toluidine blue. All DR were observed in bone tissue, especially in the bone remodeling area. Isolated OB maintained DR expression, which allowed their study in vitro. Isolated OB expressed tyrosine hydroxylase, the rate-limiting enzyme for dopamine production, and contained dopamine. The activation of D2-like DR significantly increased bone mineralization in RA osteoblasts and increased osteoclastogenesis but did not alter the expression of OC markers nor bone resorption. DR were found in the bone remodeling area of human bone tissue and dopamine can be produced by osteoblasts themselves, thus suggesting a local autocrine/paracrine pathway of dopamine in the bone. D2-like DRs are responsible for bone mineralization in osteoblasts from RA patients without an increase in bone resorption, thus suggesting the D2-like DR pathway as a possible future therapeutic target to counteract bone resorption in arthritis.

## 1. Introduction

Rheumatoid arthritis (RA) is an autoimmune disease characterized by chronic joint inflammation, articular bone erosion, and consequently, joint destruction that can lead to a complete loss of function [1]. Additionally, RA is not simply a disease of the joints but can affect many other organs and cause, for instance, systemic osteoporosis (reviewed in [2]). Besides the higher mortality associated with RA, one of the main issues is the morbidity associated with this pathology and the consequent reduced quality of life of RA patients. Joint inflammation in RA affects multiple sites causing widespread pain, and the subsequent joint destruction can lead to severe disability affecting all aspects of motor function, from walking to fine movements of the hand [3]. Moreover, work disability is one major issue in rheumatic diseases [4], with around one third of patients leaving employment prematurely [5].

Nowadays, there are many therapeutics available that can significantly reduce the impact of RA. However, bone destruction and osteoporosis remain an issue. In fact, most osteoporosis treatments aim at inhibiting bone resorption, whereas few therapies are capable of actively promoting new bone tissue formation in order to restore the bone matrix already lost. They include bisphosphonates, the anti-RANKL antibody denusomab, and parathormone-acting molecules. Besides that, such therapies are usually not prescribed in the very early stage of the disease. Bone loss, however, can be observed in RA patients with recent onset of rheumatoid arthritis [6], suggesting that the start of the destructive phase of disease may occur much earlier than previously expected, and even before the symptomatic phase. This effect can be amplified by the clinical need that glucocorticoids must frequently be given in the early phase of disease and thus contribute to bone mass loss and osteoporosis. A better understanding of the mechanisms involved in the early, systemic bone loss in RA patients is therefore required.

Bone remodeling is based on the correct balance between bone resorption by osteoclasts and bone formation by osteoblasts. The proinflammatory cytokines produced during chronic inflammation, such as IL-6 and TNF in RA, induce an uncoupling of bone formation and resorption, resulting in the activation of osteoclasts and, consequently, significant bone loss in patients with inflammatory joint diseases (reviewed in [7,8]). Additionally, macrophage migration inhibitory factor (MIF) plays a crucial role in bone metabolism in humans, as high plasma levels of MIF as well as a mutation in the MIF promoter are associated with low bone density in postmenopausal women [9,10]. Not only cytokines but also autoantibodies are described to induce osteoclastogenesis, thus corroborating bone loss (reviewed in [11]). Besides osteoclast activation, bone loss is also caused by osteoblast inhibition. For example, it has been described that B cells are enriched in the bone marrow and inhibits osteoblasts via the release of CCL3 and TNF [12], but also the abnormal activity of Wnt signaling, the receptor activator of nuclear factor-κB ligand (RANKL)-osteoprotegerin (OPG) signaling, as well as the bone morphogenetic proteins (BMPs) pathway and other mechanisms have been described to be altered in osteoblasts in rheumatic diseases [13].

In RA patients, three major forms of bone loss are described: local bone erosion in inflamed joints, systemic osteoporosis and periprosthetic aseptic osteolysis (reviewed in [14,15]). As mentioned above, systemic osteoporosis in RA patients may occur already in the early stage of the disease [6], even before systemic inflammation starts, thus suggesting a major role for antibodies against citrullinated proteins as trigger of osteoclast activation and bone erosion. Differently, local bone erosion in the inflamed joints is triggered by many cells present in the inflamed synovial tissues, such as macrophages, fibroblasts, and immune cells releasing proinflammatory mediators such as cytokines and matrix metalloproteinases, among others (revied in [14]), thus altering bone metabolism towards a bone destructive path. Another relevant form of bone loss in RA is the sometimes-occurring osteolysis in the periprosthetic region. One possible reason is the wear of prosthetic alloys, which then stimulates inflammatory response and local bone resorption (revied in [15]).

While looking for unknown pathways involved in RA bone metabolism, some clinical evidence, together with a few in vitro studies and our previous results, led us to hypothesize an involvement of the neurotransmitter dopamine in bone erosion in RA. Dopamine is a neurotransmitter of the central nervous system controlling movement, emotion, cognition, and neuroendocrine interactions. Recent evidence supports a key role of dopaminergic pathways in the modulation of immunity in the periphery [16,17,18,19,20,21,22]. Dopamine acts via five DA receptors (DRs), which are expressed in most if not all human immune cells. Five different DR were described in the late 1970s. These receptors are called D1-D5 DR and are grouped into two families: the D1-like DR and D2-like DR [23]. D1- and D5-DR belong to the D1-like DR and are coupled to Gαs, whereas D2-, D3-, and D4-DR belong to the D2-like DR [24]. Apart from the prevailing mechanism of cAMP regulation, DR can also regulate a variety of further signaling pathways [24]. During chronic inflammation, DR can switch the cytoplasmatic subunit from Gαs to Gαi signaling [25]. Furthermore, DR can exist in oligomeric form [22,26]. In addition to expressing DR on their membrane, immune cells are able to synthesize and utilize dopamine as an autocrine/paracrine transmitter [27,28]. The immune modulatory effects of DA most likely depend on its concentration, the presence of specific DR subtype(s), and the specific cell types involved.

Some clinical evidence suggests the involvement of DA in RA: the incidence of RA in schizophrenia patients is much lower than in the general population, and a possible protective effect of DA antagonists prescribed to schizophrenia patients was suggested [29]. Moreover, patients affected by Parkinson’s disease have a high incidence of RA, bone fracture and osteoporosis, and a possible role of dopamine unbalance was hypothesized [30,31]. However, the relative effect of corticosteroid and DA on bone resorption in Parkinson’s patients is controversial. Furthermore, in RA patients, a local high concentration of DA was measured in the synovial fluid, and an increased presence of DRs in cartilage from osteoarthritis (OA) patients was observed compared to healthy controls [32]. Additionally, restless leg syndrome is often described in RA patients, but the effects of l-DOPA treatment on bone resorption are not described.

Besides this clinical evidence, the involvement of dopamine in RA was already demonstrated in vitro, as well as in animal models of arthritis. Previous studies showed the efficacy of SCH23390, a D1-like DR antagonist, in reducing the severity of collagen-induced arthritis (CIA) in mice [33]. We previously showed that synovial cells of RA patients produce dopamine [34] and that RA synovial fibroblasts (RASF) and macrophages express high amount of DRs [35]. Locally produced DA can thus affect synovial DRs in an autocrine/paracrine manner. In fact, the DA treatment of RASF strongly reduced IL-6 and IL-8 release in RASF from patients not treated with biologics [35] but induced migration of synovial fibroblasts from long-treated RA patients [36]. Taken together, this evidence demonstrates the involvement of DA on RA synovitis. A possible influence of DA on bone remodeling is also plausible, although there are only few studies suggesting a direct link between DA and bone metabolism. In one study, D2-like receptor signaling was shown to inhibit osteoclastogenesis in vitro, therefore implying a protective effect on bone resorption [37]. In another study, D1-like receptor antagonist SCH-23390 reduced murine osteoclast differentiation in vitro and bone destruction in a CIA mouse model [33].

Despite these interesting findings and some clinical evidence available as described above, to the best of our knowledge, the role of dopamine and DRs on bone metabolism in RA patients has not yet been described. We therefore performed preliminary experiments to evaluate the presence of DRs in the bone and joint interface of RA patients and their functionality. We hypothesize that the locally activated dopaminergic pathway in the inflamed joints also influences bone metabolism in RA, and that unraveling dopaminergic pathways on bone metabolism can open the way to new therapeutic approach for controlling bone resorption in RA.

## 2. Materials and Methods

### 2.1. Study Cohort

Bone samples were obtained from RA (*n* = 14) and OA (*n* = 14) patients undergoing joint replacement surgery (Dept. of Orthopaedics and Trauma Surgery, Markus-Hospital, Frankfurt, Frankfurt/Main, Germany; Dept. of Orthopedics and Orthopaedic Surgery, University Hospital Giessen, Giessen, Germany) (see Table 1 for patient characteristics). As shown in Table 1, patients’ age at surgery as well as the gender distribution differ between the two groups. These differences reflect the demographic features of these two cohorts, as OA normally has a later onset compared to RA, and RA affects more women than men [38,39]. All specimens were obtained with the approval of the Ethics Committee of the Justus-Liebig-University Giessen (approval number 74/05, 66/08). Patients gave written informed consent and fulfilled the criteria of the American College of Rheumatology.

For osteoclast differentiation, peripheral blood mononuclear cells (PBMCs) were isolated from blood samples of patients with confirmed diagnosis of RA (*n* = 9) from the Rheumazentrum Ruhrgebiet (Herne, Germany) and healthy controls (HC, *n* = 8) (See Table 1 for subject characteristics). The study was approved by the ethics committee of IfADo (IfADo2017/125/2019-02-04). All subjects signed informed consent for study participation, and peripheral venous blood was collected in Li-Heparin tubes (Sarstedt, Nümbrecht, Germany) and processed within 6 h upon blood collection.

### 2.2. Bone Tissue Specimens and Osteoblast Cell Culture

Bone tissue preparation for the histological analysis was performed as previously described [40]. Briefly, bone tissue was fixed in 4% formaldehyde (Carl Roth, Karlsruhe, Germany) for 24 h and then decalcified in 20% sodium EDTA (Sigma Aldrich, Taufkirchen, Germany) for 6 weeks. Bone tissue was embedded in paraffin, and 5 µm sections were prepared and deparaffinized for tissue stainings and immunohistochemistry.

For in vitro cell culture, osteoblasts were isolated and cultured as previously described [40]. Briefly, bone samples were minced to approximately 2 mm^2^ fragments and washed. Bone fragments were then placed in culture flasks containing alphaMEM Glutamax (Gibco, Life technologies, Darmstadt, Germany) supplemented with 10% FCS, 100 U/mL penicillin (Life technologies, Darmstadt, Germany), and 10 μg/mL streptomycin (Life technologies, Darmstadt, Germany) and cultured at 37 °C, 5% CO_2_. After cells grew out and reached confluency, the bone fragments were removed, and the cells were passaged using trypsin. For simplification, isolated cells are mentioned as osteoblasts (OB), but it should be considered that cells isolated with this protocol are not pure osteoblasts, and they can also include other mesenchymal cells and osteoblast precursors. Nevertheless, the expression of osteocalcin was detected in the majority of cells in all analyzed patients (OA *n* = 6, RA *n* = 4, Appendix A). Experiments were performed with OB cultured up to passage 4.

### 2.3. Osteoclast Isolation and Culture

Peripheral blood mononuclear cells (PBMCs) were isolated by Ficoll density gradient centrifugation (density: 1.077 g/mL; PanBiotech, Aidenbach, Germany) and washed twice with DPBS (Gibco).

Cells were then resuspended in alphaMEM Glutamax (Gibco, Life technologies, Darmstadt, Germany) supplemented with 10% FCS, 100 U/mL penicillin (Life technologies, Darmstadt, Germany), 10 μg/mL streptomycin (Life technologies), and 25 ng/mL MCSF (BioLegend, Amsterdam The Netherlands). The cells were seeded at specific concentrations, depending on the culture plate used (1.5 × 10^6^ cells/well in 6 well plates, 0.15 × 10^6^ cells/well in 48 well plates, and 0.05 × 10^6^ cells/well in 96 well plates) and cultured at 37 °C, 5% CO_2_. After 24 h, the medium was replaced with a differentiation medium containing alphaMEM Glutamax (Gibco, Life technologies, Darmstadt, Germany) supplemented with 10% FCS, 100 U/mL penicillin (Life technologies), 10 μg/mL streptomycin (Life technologies, Darmstadt, Germany), 25 ng/mL MCSF, and 50 ng/mL RANKL (BioLegend, Amsterdam The Netherlands). As an osteoclast differentiation control, cells were cultured without RANKL. The cells were cultured for 14 days, and the medium was replaced twice a week.

### 2.4. Immunohistochemistry of Dopaminergic Receptors in Bone Tissue

Bone slices were firstly deparaffinized and re-hydrated. Afterwards, antigen retrieval was performed by enzymatic digestion with proteinase K (Sigma Aldrich, Taufkirchen, Germany, 0.6 U/mL). Endogenous peroxidases were inactivated by incubation with 3% H_2_O_2_ solution, and unspecific binding sites were blocked with phosphate-buffered saline (PBS) containing 10% fetal calf serum, 10% chicken serum, and 10% albumin (Sigma Aldrich). Afterwards, samples were incubated overnight at 4°C with the respective primary antibody (Appendix A) and then counterstained by using Histofine (Nichirei Biosciences, Tokyo, Japan) and AEC substrate chromogen (Dako), as indicated by the manufacturers. Control staining with the secondary antibody alone (Histofine) was carried out in parallel and showed no positive staining.

### 2.5. Flow Cytometry

Cells were cultured in 75 cm^2^ flasks. At 80% confluence, cells were detached using accutase (BioLegend, Amsterdam The Netherlands) and counted. D1DR, D3DR, and D5DR were stained extracellularly, whereas antibodies against D2DR and D4DR bind to intracellular domain of the respective DR; thus, an intracellular staining was required. The antibodies used are listed in Appendix A.

#### 2.5.1. Extracellular Staining

For each staining condition, around 100,000 cells/well were dispensed in a 96-well V-bottom plate and kept on ice during the whole staining procedure. First, cells were incubated with specific antibodies in FACS buffer (2% FBS in PBS) for 20 min at 4 °C. After a washing step with FACS buffer, the cells were immediately analyzed.

#### 2.5.2. Intracellular Staining

For intracellular staining, cells were fixed with FACS fixation buffer, consisting of 2% formaldehyde (Carl Roth, Karlsruhe, Germany) in FACS buffer, for 10 min at RT. Afterwards, the cells were washed with FACS buffer and then permeabilized with FACS Permeabilizing Solution 2 (Becton, Dickinson, BD, Heidelberg, Germany) for 10 min at RT. The cells were washed again with FACS buffer and then stained intracellularly for 20 min at RT. After another washing step with FACS buffer, unconjugated primary antibodies were labeled with a secondary PE-labeled donkey anti-rabbit antibody during 20 min incubation at RT. Finally, the cells were washed with FACS buffer and immediately analyzed.

The samples were analyzed on a BD LSR Fortessa. At least 10,000 events were measured. The data were analyzed with FlowJo Version 10.3 (FlowJo LLC, Becton Dickinson, Ashland, OR, USA).

### 2.6. Dopamine Quantification via ELISA

For the quantification of dopamine content in primary osteoblasts, 1 × 10^6^ cells were pelleted by centrifugation at 400× *g*, 4 °C for 10 min. Dry pellets were then frozen at −80 °C. TriCat ELISA was performed as indicated by the manufacturer (IBL, Tecan, Männedorf, Switzerland). Briefly, cell pellets were lysed in 100 μL 0.1 M HClO_4_ with 100 μM ascorbic acid (Sigma Aldrich, Taufkirchen, Germany). Thereafter, catecholamines were extracted and analyzed as described by the manufacturer. Limit of detection for dopamine was 4 pg/mL.

### 2.7. Mineralization Assay

Osteoblasts were seeded in a 24-well plate (10 × 10^3^ cells/well) and cultured until reaching 90% confluence. Afterwards, the culture medium was replaced with a mineralization medium (complete cell culture medium +50 µg/mL L-ascorbic acid, 5 mM ß-glycerophosphate and 10 nM dexamethasone, from Sigma Aldrich Carl Roth and Enzo Life Sciences, respectively). The cells were cultured for 21 days, and the medium was replaced twice a week. As control, cells were cultured with culture medium without mineralization additives.

Afterwards, cell layers were washed once with PBS and fixed with 4% formaldehyde (Carl Roth, Karlsruhe, Germany) in PBS for 10 min. After washing the monolayer again with PBS, cells were stained with 40 mM alizarin red solution (Merck, Darmstadt, Germany) for 30 min at room temperature. After two wash steps with distilled water, the monolayer was air-dried overnight. A visualization of the matrix production was performed using light microscopy. Colorimetric detection and quantification was performed by acetic acid extraction, as previously described [41].

### 2.8. Osteoblast Stimulation with DR Agonists

To study the effect of dopamine on primary osteoblasts, cells were seeded in a 24-well plate (10 × 10^3^ cells/well), cultured until they reached 80–90% confluence in complete culture medium and then treated with specific agonists of D1-like receptors (Fenoldopam, Tocris Bioscience, Wiesbaden-Nordenstadt, Germany) or D2-like receptors (Ropinirole, Tocris Bioscience, Wiesbaden-Nordenstadt, Germany) at different concentrations for 24 h. Afterwards, supernatants were collected and frozen in aliquots at −80 °C until analysis.

To study the effect of dopamine on activated osteoblasts, cells were cultured following the mineralization protocols described above. After 3 weeks, cells were treated with dopamine agonists in fresh medium for further 24 h. Afterwards, supernatants were collected and frozen in aliquots at −80 °C until analysis.

The concentrations chosen for Fenoldopam and Ropinirole were around the concentration described to have the best binding affinity, ±5–10× and had no toxic effects on the cells.

### 2.9. Cytokine Quantification via ELISA

The quantification of MIF, IL-6, and IL-8 was performed as described by the manufacturer (BioLegend, Amsterdam The Netherlands). Samples were measured in duplicates or in single detection depending on sample availability. Optimal dilutions of supernatants were determined in preliminary assays.

### 2.10. Quantification of Differentiated Osteoclasts via TRAP Staining

In order to find out the effect of dopamine on osteoclast differentiation, PBMCs were cultured in 96-well plates with osteoclast differentiation medium for two weeks, as described above, in the presence or not of the D1-like DR agonist fenoldopam and the D2-like DR agonist ropinirole at different concentrations. Fresh agonists were added to the cell culture at each medium change. After 14 days, cells were washed once with DPBS and fixed in 4% formaldehyde. After washing with DPBS, the fixed cells were stained for tartrate resistant acidic phosphatase (TRAP) (Sigma-Aldrich, Taufkirchen, Germany). The staining was conducted as described by the manufacturer. Eight pictures per well were taken at a magnification of 100×. For this assay, osteoclasts were defined as TRAP-positive cells with more than two nuclei.

### 2.11. RNA Isolation and Real-Time PCR

PBMC were culture in 6-well plates with osteoclast differentiation medium for 14 days, as described above, in the presence or not of the DR-specific agonists fenoldopam and ropinirole. Afterwards, cells were washed in DPBS, lysed in Trizol (Invitrogen, Karlsruhe, Germany), and then stored at −80 °C. RNA extraction was performed following manufacturer’s instructions. Then, 200 ng of RNA from each condition was used for reverse transcription (Qiagen RT kit, Qiagen, Hilden, Germany). The expression of genes involved in osteoclast differentiation (*NFATC1*) and function (*ACP5* or TRAP, *CTSK* and *MMP9*) was analyzed by real-time PCR using a SYBR green master mix (Sso advanced, BioRad, Feldkirchen, Germany) and specific primers, as indicated in Appendix A. The expression of glyceraldehyde 3- phosphate dehydrogenase (*GAPDH*) was used as reference. PCR were conducted with a CFX thermocycler (BioRad, Feldkirchen, Germany). All PCR were performed for 40 cycles with two-temperature protocols: after a first 120″ denaturation step at 95 °C, each cycle included a 5″ denaturation and 30″ annealing/elongatin step. For TRAP and NFATc1, the annealing/elongation temperature was 56 °C, and for MMP9 and CTSK, it was 60 °C.

### 2.12. Bone Resorption

Isolated PBMCs were cultured with osteoclast differentiation medium on top of bone slices (Ids) in 48 well plates for 14 days, in addition or not of the specific DR agonists fenoldopam and ropinirole at different concentrations as described above. Cells were then removed by washing bone slides with distilled water and then with sonication. The resorption pits were stained with 1% toluidine blue (Carl Roth, Karlsruhe, Germany) in water for 1 min and rinsed with distilled water 5 times. Pictures of the bone slices were taken at a magnification of 100×. The resorption area was quantified in three representative pictures using Photoshop.

### 2.13. Statistical Analysis

Statistical analysis and graphic design were performed with Prism 9 software (Version 9.3.1, GraphPad Software, San Diego CA). The numbers of investigated subjects are indicated in the figures for each experiment. Raw data were used for statistical analysis with Wilcoxon matched pairs signed rank test to compare treatments versus untreated samples of the same subject. To compare basal raw value within groups, the Mann–Whitney test was used.

Single dots represent the mean value of individual patients. Histograms show mean and SEM.

## 3. Results

### 3.1. Dopamine Receptors Are Expressed in the Bone of OA and RA Patients

All dopamine receptors were detected in the bone tissue of OA and RA patients (Figure 1), whereas the overall expression of DR was higher in RA compared to OA. D1DR and D4DR expression was comparable in OA and RA patients and were detected both in osteoclasts and osteoblasts. The expression of D2DR and D3DR was very weak in OA and stronger in RA, where the receptors were detected both in osteoclasts and osteoblasts. D5DR expression was weak in both patient cohorts.

### 3.2. Dopamine Receptors Are Expressed in Isolated Osteoblasts

Isolated osteoblasts in culture maintained DR expression, as detected via FACS analysis (Figure 2). No significant differences between OA and RA were observed. The expression of D5DR was extremely weak, thus confirming the histological staining, and the expression of D2-like DR was overall stronger than D1-like DR expression.

### 3.3. Osteoblasts Synthesize Dopamine

Osteoblasts were detected positive for tyrosine hydroxylase (TH), the rate limiting enzyme for dopamine synthesis (Figure 3A), and dopamine was measured in isolated osteoblasts (Figure 3B), thus suggesting that osteoblasts can synthesize dopamine and use it in an autocrine/paracrine way. The total amount of dopamine tended to be higher in OA compared to RA patients (Figure 3B), even though no significant difference was reached.

### 3.4. D2-like DR Stimulation Affects Mineralization

Cell culture with mineralization medium led to bone matrix formation after 21 days (Figure 4A). In OA patients, mineralization tended to be higher than in RA patients, and a significant variation within patients was observed (Figure 4B). Treatment with ropinirole, a D2-like specific agonist, led to a significant increase in bone mineralization, specifically only in RA (Figure 4C) at the highest concentration of ropinirole, whereas no differences were observed in OA patients. D1-like stimulation only slightly increased bone matrix formation in RA. There was a strong variation of the overall mineralization obtained in each patient (Figure 4B), probably due to the inflammatory state or the treatments at time of surgical intervention or to the difference in osteoblast amount in the isolated cells from each subject. Nevertheless, the increase in mineralization after D2-like DR stimulation was detected in all RA patients at the two highest concentrations of D2-like DR agonist, thus suggesting a direct influence of the dopaminergic pathway on bone matrix formation.

### 3.5. DR Stimulation Increases MIF and RANKL Release

Primary osteoblasts isolated from OA and RA patients released MIF, RANKL, IL-8, and IL-6, and no differences were observed between the two cohorts (Figure 5A,C,E,G). The stimulation of D2-like DR led to a significant increase in MIF release in RA osteoblasts at the highest concentration of D2-like agonist, whereas MIF release tended to increase in OA OB as well, but no significant differences were observed (Figure 5B). The stimulation of D1-like DR led to a slight but not significant increase in MIF in RA OB, and no differences were observed in OA OB (Figure 5B). The activation of D1-like and D2-like DR significantly increased the release of RANKL, but stronger and dose-dependent effects were observed in RA patients compared to OA (Figure 5D). IL-8 release was slightly reduced in cells from OA and increased in cells from RA after D2-like DR stimulation, but no significant differences were observed (Figure 5F). Similar results were also observed for IL-6, with a slight but not significant increased release in osteoblasts from RA after D2-like stimulation, and no differences in OA OB (Figure 5H).

To find out if dopamine also influences the cytokine release of activated osteoblasts, we firstly induced mineralization for 21 days as described above, and then stimulated DR for 24 h. Our results demonstrate a similar amount of MIF, RANKL, IL-8, and IL-6 release in OA and RA osteoblasts (Figure 6A,C,E,G). DR stimulation did not affect the release of MIF (Figure 6B), RANKL (Figure 6D), IL-8 (Figure 6F), and IL-6 (Figure 6H) in both groups.

### 3.6. DR Stimulation Affects Osteoclast Differentiation

A culture of PBMC with osteoclast differentiation medium for 14 days induced osteoclast differentiation (Figure 7A,B). The stimulation of D2-like DR with ropinirole 10^−6^ M during differentiation significantly increased osteoclast differentiation in RA, whereas D1-like DR stimulation tended to inhibit osteoclastogenesis (Figure 7D). No significant effects of DR stimulation were observed in HC (Figure 7C).

### 3.7. DR Stimulation Does Not Alter Osteoclast Function

In order to find out if DR stimulation also affects osteoclast function, we quantified the expression of tartrate-resistant phosphatase (TRAP), nuclear factor of activated T-cells, cytoplasmic 1 (NFATc1), cathepsin K (CTSK), and matrix metalloprotease 9 (MMP9), which are described to be involved in osteoclastogenesis and the osteoclast function.

Our results revealed a slight but not significant increase in TRAP expression after D2-like DR stimulation (Figure 8A), thus confirming the results observed with the TRAP staining above. NFATc1 expression was inhibited in RA OC after D1-like and D2-like DR stimulation (Figure 8B), whereas the effects observed were very small and probably not of physiological relevance. No significant effects of DR stimulation were observed for CTSK (Figure 8C) and MMP9 (Figure 8D) expression.

To further investigate possible effects of DR-stimulation on OC function, we also investigated bone resorption (Figure 9A,B). Osteoclasts differentiated from RA PBMCs induced significantly more bone resorption than osteoclasts differentiated from HC subjects (Figure 9C), but DR stimulation had no effects on bone resorption in both cohorts (Figure 9D).

## 4. Discussion

Systemic osteoporosis still represents a big challenge in the treatment of RA, due to the early onset and the associated morbidity [6]. In this manuscript, we evaluated the DR-related pathway involved in bone metabolism in RA patients.

Our results demonstrate the presence of DR in bone tissue in human subjects. Previous studies described the expression of DR in human bone mesenchymal stem cells and an involvement of D1-like DR in osteogenic differentiation [42]. The expression of DR and the involvement of dopamine on bone metabolism were also suggested in rat osteoblasts [43], as well as in mice [44,45]. Despite these interesting results, to the best of our knowledge, the presence of DR in human bone tissue has not been investigated in detail, although clinical evidence already suggests a role for dopamine on bone metabolism also in humans.

The presence of DR both in OA as well as in RA patients suggests a role for dopamine also in the physiology of human bone metabolism. Considering that dopamine is not just a neurotransmitter but has a physiological role on many cells outside of the nervous system, such as on immune cells [16,17,18,19,20,21,22], it is plausible that dopamine plays a role also in physiological bone remodeling. Moreover, immune cells are able to produce and release dopamine [27,28] and could therefore release dopamine in the bone marrow.

Histological staining revealed an overall stronger expression of DR in RA bone compared to OA, thus suggesting an activation of the dopaminergic pathway in RA. However, there seems not to be any significant difference in the DR expression in isolated osteoblasts, thus suggesting that DR expression in the bone may vary depending on the bone region, with stronger differences in the bone remodeling areas but no stronger differences elsewhere. Our results revealed the expression of DR also in osteoclasts, thus suggesting an involvement of dopamine also on bone degradation.

We could also demonstrate the presence of dopamine in isolated osteoblasts, thus suggesting that dopamine can be synthesized and used in a paracrine/autocrine manner in the bone. Of interest, the amount of dopamine tended to be lower in osteoblasts from RA patients compared to OA patients. This difference in the dopamine amount could be involved in the lower bone formation in RA. Indeed, our results suggest that the stimulation of D2-like DR increased bone formation. Of interest, despite the similar amount of DR expression on isolated osteoblasts in OA and RA patients, D2-like stimulation did not affect mineralization in OA, thus suggesting that the proinflammatory milieu in RA patients and/or biological treatments are responsible for the different responsiveness of osteoblasts to D2-like DR activation in the two groups. Indeed, previous publications suggest that biological treatments can modulate DR [46], and chronic inflammation in RA can lead to a switch of the G_alpha_ subunit of GPCR [25], thus possibly altering the effect of GPCR such as DR. These results point out the importance of studies performed on primary osteoblasts isolated from the bone tissue in order to reproduce a more truthful experimental setting and to obtain clinically relevant readouts, despite the experimental challenge of osteoblast isolation from the bone.

In order to find out if dopamine only modulates bone formation or also regulates the release of cytokines involved in bone metabolism, we measured the concentration of different cytokines involved in bone remodeling, such as MIF, RANKL, IL-6, IL-8, and TNF. The amount of TNF was very low and, in most cases, below the detection limit; therefore, no data could be shown. All other cytokines were present at comparable levels in OA and RA osteoblast culture. The stimulation of D2-like DR in primary osteoblasts led to an increase in MIF at the highest concentration of agonist in RA, and RANKL was increased both in OA and RA after D1-like and D2-like stimulation, thus suggesting an effect of dopamine not only on bone formation but also on osteoclast differentiation. No dopaminergic effects were observed on IL-8 and IL-6, and the stimulation of activated osteoblasts with DR agonists did not affect cytokine release at all. This could be due to a different expression level of DR in primary and in activated osteoblasts. Due to the limited amount of cells available, it was not possible to prove this hypothesis in the current study. These findings suggest that D2-like stimulation could increase mineralization, which would make the D2-like DR stimulation in osteoblast an interesting therapeutic target against bone loss specifically in RA, but the increase in MIF and RANKL also suggest an effect on osteoclasts that should be considered.

To investigate possible direct effects of dopamine on osteoclastogenesis and bone resorption, in vitro experiments on osteoclasts were also performed and revealed that the stimulation of D2-like DR induces osteoclastogenesis in RA but has no effects on osteoclast activation and bone resorption. Therefore, our results point out the therapeutic potential of D2-like DR stimulation in the bone to achieve bone formation and counteract osteoporosis in RA.

One limitation of this study is the strong variation within subjects, as is currently known for primary cells and revealed by our results. This is due to differences in age, sex, therapy, and inflammation state within the patients. Additionally, cells isolated from bone sample are not solely osteoblasts, but mesenchymal cells and osteoblast precursors could also be present, thus increasing variation. Another limitation of the study was the small number of osteoblasts available for each patient, which did not allow us to perform more experiments. However, we are convinced that using primary osteoblasts isolated from bone tissues leads to more clinically relevant results. Despite the variation within patients, we could already observe significant effects of dopamine, which suggest a relevant involvement of the dopaminergic pathway on bone metabolism.

## 5. Conclusions

Taken together, our results suggest an involvement of D2-like DR pathway on bone formation in primary osteoblasts from RA patients and the physiological presence of DR in the human bone. Based on these results, an involvement of the dopaminergic pathway in bone metabolism in other skeletal disorders is also plausible and should be investigated.

These results do not only suggest the dopaminergic pathway as a therapeutic target for inducing bone formation, but also point out the possible side-effects on bone density of dopaminergic drugs used in RA patients because of comorbidity, which should be further investigated and taken under consideration.

## Figures and Tables

**Figure 1 cells-11-01609-f001:**
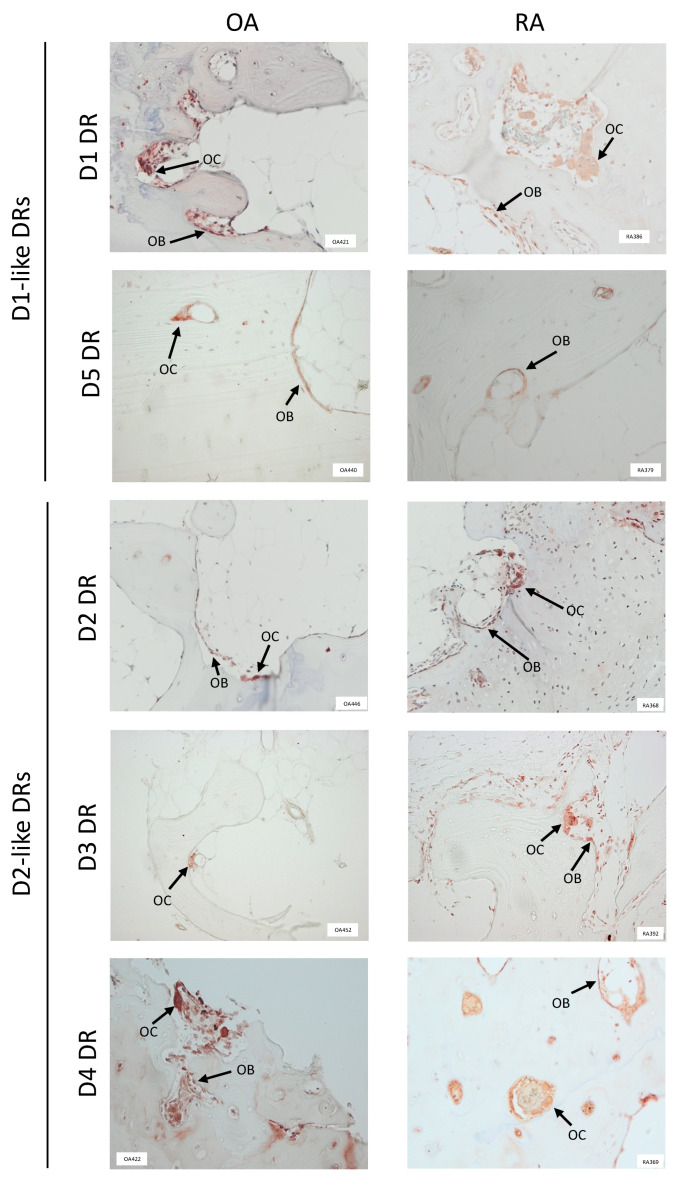
Dopamine receptors are expressed in human bone tissue. Dopamine receptors (DR) were stained in paraffin embedded, decalcified bone samples from at least 10 patients for each group. Representative pictures showing DR positivity at the border of newly formed, non mineralized bone are shown. OB = osteoblasts; OC = osteoclasts. 100× magnification was used in all pictures.

**Figure 2 cells-11-01609-f002:**
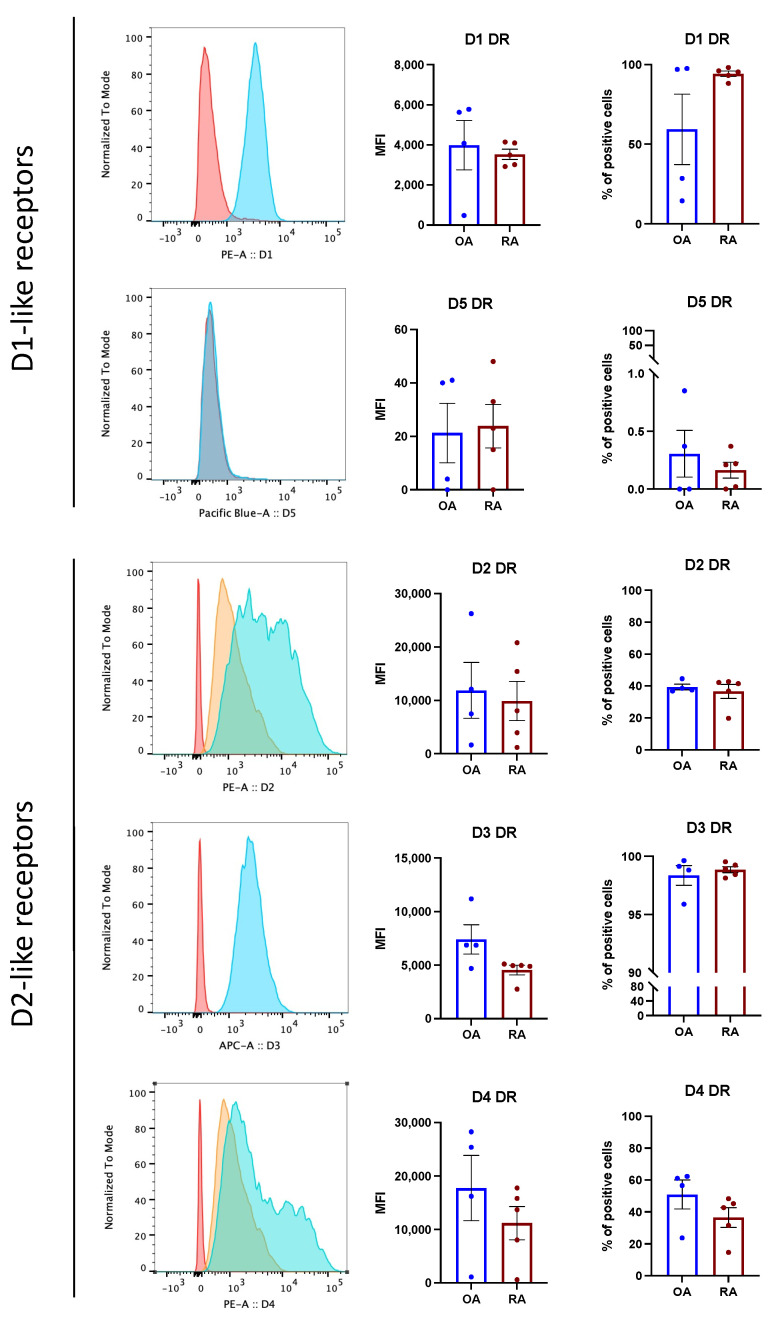
Primary osteoblasts express DR in vitro. Osteoblast were stained for DR by flow cytometry at passage 3. (**Left**): Representative histograms showing DR expression in RA. Red: unstained control; orange: secondary antibody alone (only for unconjugated primary antibodies); blue: specific DR staining. (**Right**): Mean fluorescence intensity (MFI) and % of positive DR stained cells in OA (blue bars) and RA (red bars). Histograms represent mean ± SEM. Each point represents one patient. OA *n* = 4, RA *n* = 5.

**Figure 3 cells-11-01609-f003:**
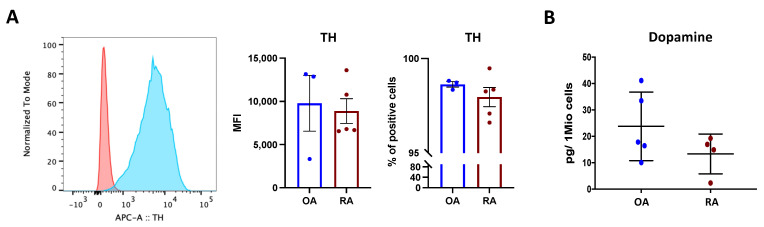
Dopamine synthesis in isolated osteoblasts. Tyrosine hydroxylase (TH), the rate-limiting enzyme for dopamine synthesis, was quantified in isolated osteoblasts at passage 3. (**A**): Representative histograms (**left**) showing TH expression in RA. Red: unstained control; blue: TH staining. Right: mean fluorescence intensity (MFI) and % of positive TH-stained cells in OA (blue bars) and RA (red bars). Histograms represent mean ± SEM. Each point represents one patient. OA *n* = 4, RA *n* = 5. (**B**): Quantification of dopamine content in cell lysates at passage 3, indicated as picogramm (pg) in one million (Mio) cells. Each point represents the mean value of duplicate measurements of one patient. OA *n* = 5, RA *n* = 4.

**Figure 4 cells-11-01609-f004:**
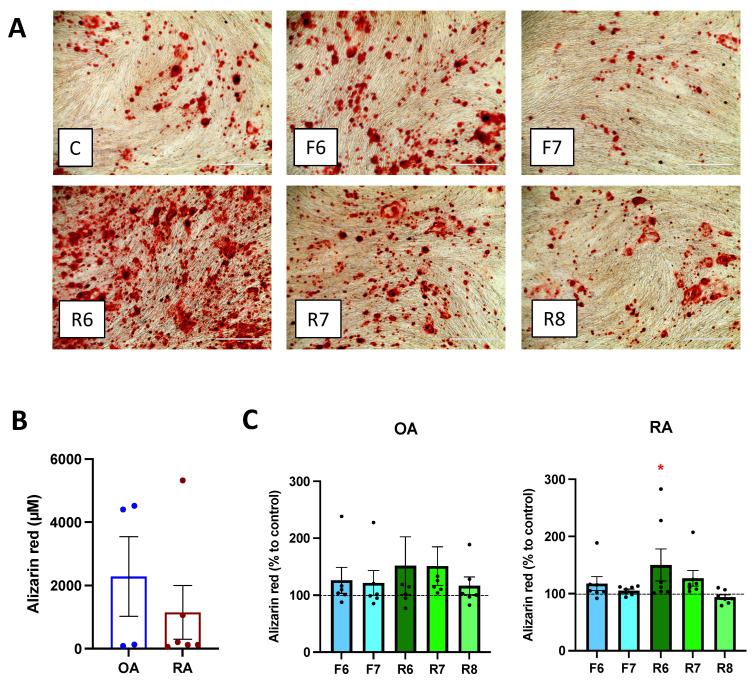
Dopamine affects mineralization. Isolated osteoblasts were cultured for 3 weeks with mineralization medium and specific DR agonists prior alizarin red staining of calcium deposits. (**A**): Microscopy picture of the stained wells. Magnification 100×, scale bar 400 μm. (**B**,**C**): After visualization of the matrix, Alizarin red staining was quantified by acetic acid extraction. (**B**): Basal level of Alizarin red amount in OA and RA patients. CD: Results were normalized to control without DR agonist treatment. OA *n* = 7, RA *n* = 9. C = control; F = Fenoldopam; R = Ropinirole; 6 = 10^−6^ M; 7 = 10^−7^ M; 8 = 10^−8^ M. Histograms represent mean + SEM. Each point represents the mean value of duplicate measurements of one patient. Raw data were used for statistical analysis with Wilcoxon matched pairs signed rank test. * = *p* < 0.05.

**Figure 5 cells-11-01609-f005:**
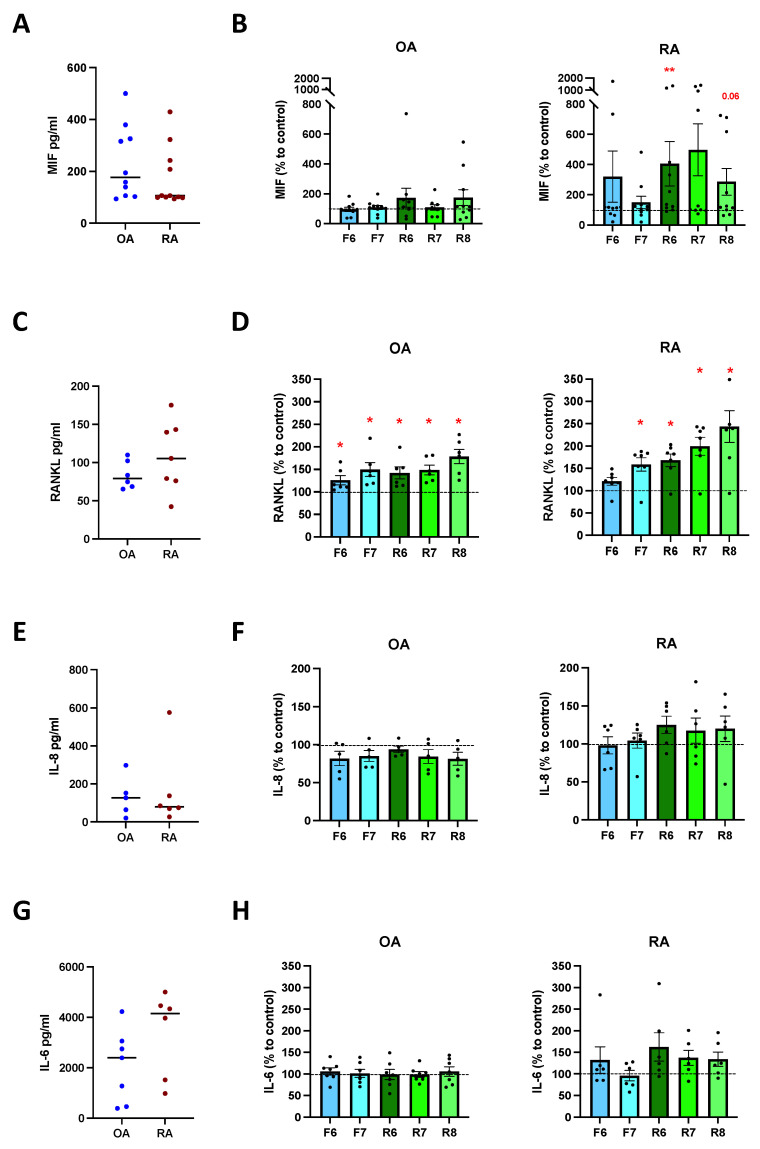
Cytokine release after DR stimulation of primary osteoblasts. Isolated osteoblasts were cultured for 24 h in presence of specific DR agonists, and cytokines were then measured in culture supernatants. (**A**): basal amount of macrophage migration inhibitory factor (MIF) in OA and RA patients. (**B**): quantification of MIF in OA (**left**) and RA (**right**) patients expressed as percentage to control without DR agonist. (**C**): basal amount of receptor activator of nuclear factor-κB ligand (RANKL) in OA and RA patients. (**D**): quantification of RANKL in OA (**left**) and RA (**right**) patients expressed as percentage to control without DR agonist. (**E**): basal amount of interleukin 8 (IL-8) in OA and RA patients. (**F**): quantification of IL-8 in OA (**left**) and RA (**right**) patients expressed as percentage to control without DR agonist. (**G**): basal amount of interleukin 6 (IL-6) in OA and RA patients. (**H**): quantification of IL-6 in OA (**left**) and RA (**right**) patients expressed as percentage to control without DR agonist. OA *n* = 4–10, RA *n* = 3–11. C = control; F = Fenoldopam; R = Ropinirole; 6 = 10^−6^ M; 7 = 10^−7^ M; 8 = 10^−8^ M. Single dots represent mean value of individual patients. Histograms show mean and SEM. Raw data were used for statistical analysis with Wilcoxon matched pairs signed rank test. * = *p* < 0.05, ** = *p* < 0.005. *p*-value near to significance (*p* = 0.06) are indicated in the figure.

**Figure 6 cells-11-01609-f006:**
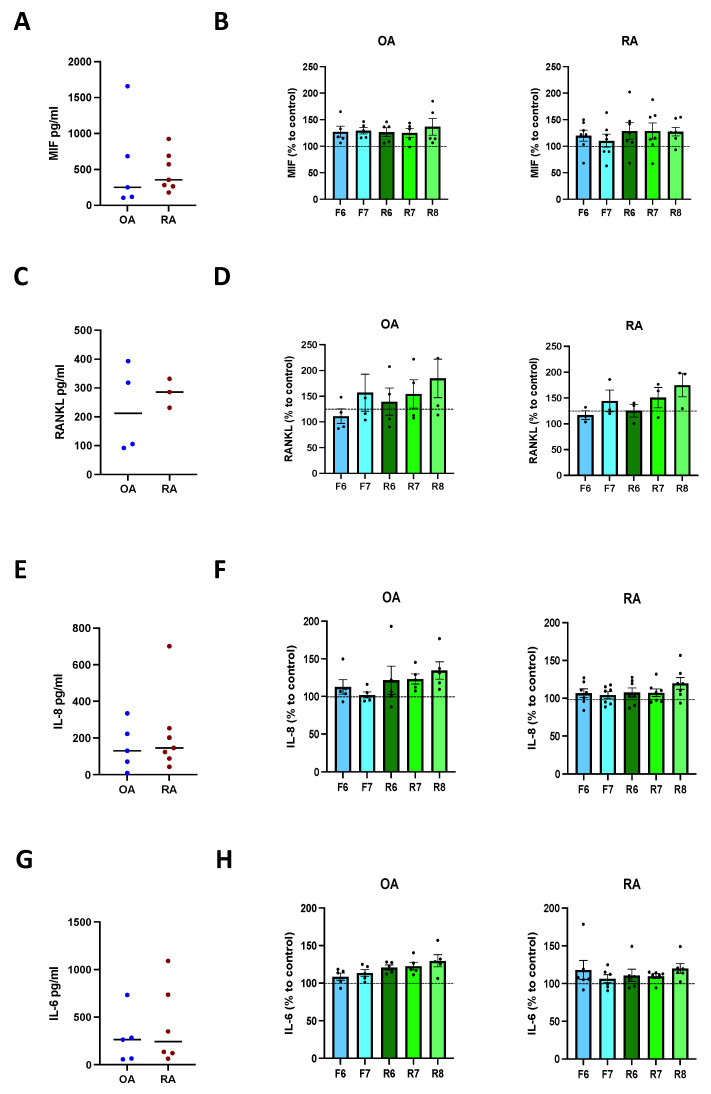
Cytokine release after DR stimulation of activated osteoblasts. Isolated osteoblasts were cultured for 3 weeks with mineralization medium, and afterwards for further 24 h in presence of specific DR agonists. Afterwards, cytokines were measured in culture supernatants. (**A**): basal amount of macrophage migration inhibitory factor (MIF) in OA and RA patients. (**B**): quantification of MIF in OA (**left**) and RA (**right**) patients expressed as percentage to control without DR agonist. (**C**): basal amount of receptor activator of nuclear factor-κB ligand (RANKL) in OA and RA patients. (**D**): quantification of RANKL in OA (**left**) and RA (**right**) patients expressed as percentage to control without DR agonist. (**E**): basal amount of interleukin 8 (IL-8) in OA and RA patients. (**F**): quantification of IL-8 in OA (**left**) and RA (**right**) patients expressed as percentage to control without DR agonist. (**G**): basal amount of interleukin 6 (IL-6) in OA and RA patients. (**H**): quantification of IL-6 in OA (**left**) and RA (**right**) patients expressed as percentage to control without DR agonist. OA *n* = 4–5, RA *n* = 3–7. C = control; F = Fenoldopam; R = Ropinirole; 6 = 10^−6^ M; 7 = 10^−7^ M; 8 = 10^−8^ M. Single dots represent mean value of individual patients. Histograms show mean and SEM. Raw data were used for statistical analysis with Wilcoxon matched pairs signed rank test.

**Figure 7 cells-11-01609-f007:**
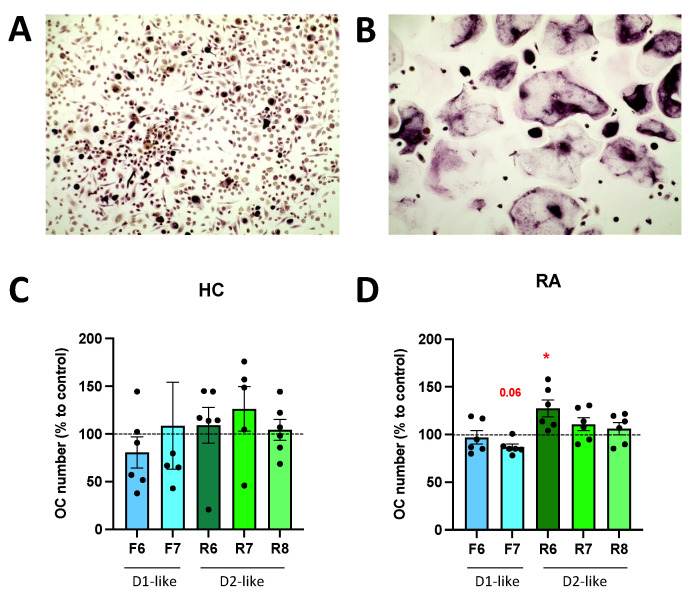
DR stimulation affects osteoclastogenesis. PBMCs were cultured with MCSF alone (**A**) or MCSF and RANKL (**B**) for 14 days and then stained for TRAP. TRAP-positive cells with more than two nuclei were counted as osteoclasts. (**A**,**B**): representative pictures from one healthy control. Magnification 100×. (**C**,**D**): quantification of osteoclast after specific DR stimulation for 14 days during OC differentiation in HC (**C**) and RA (**D**) samples. Values are expressed as percentage to control without DR agonist. HC *n* = 6, RA *n* = 6. F = Fenoldopam; R = Ropinirole; 6 = 10^−6^ M; 7 = 10^−7^ M; 8 = 10^−8^ M. Single dots represent individual patients. Histograms show mean and SEM. Raw data were used for statistical analysis with Wilcoxon matched pairs signed rank test. * = *p* < 0.05. *p*-value near to significance (*p* = 0.06) are indicated in the figure.

**Figure 8 cells-11-01609-f008:**
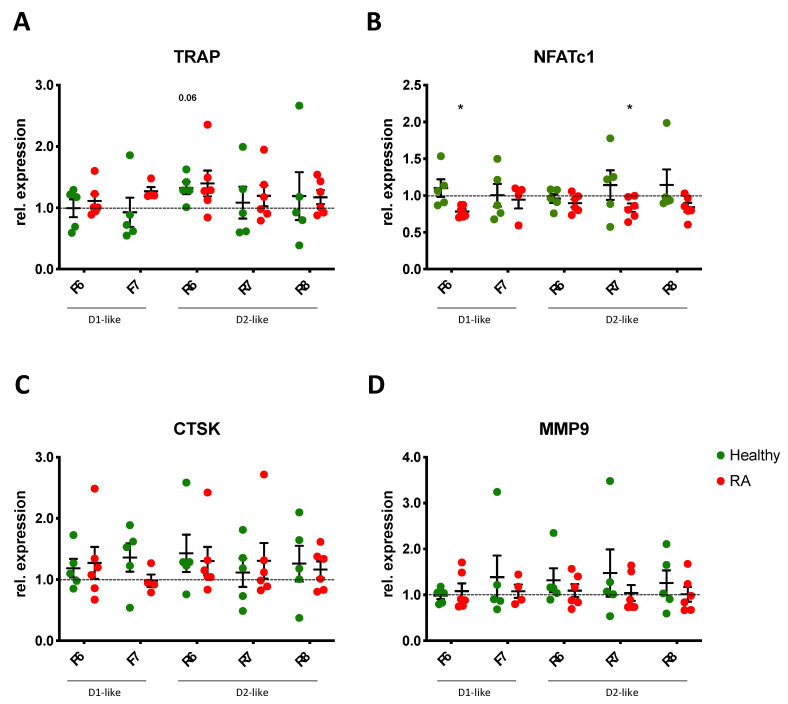
DR stimulation does not alter gene expression of OC markers. Expression of tartrate resistant phosphatase (TRAP, (**A**)), nuclear factor of activated T-cells, cytoplasmic 1 (NFATc1, (**B**)), cathepsin K (CTSK, (**C**)) and matrix metalloprotease 9 (MMP9, (**D**)) were analyzed by real-time PCR in PBMCs of healthy controls (green dots) and RA patients (red dots) cultured with OC differentiation medium for 14 days in the presence or not of DR agonists. Values are normalized to GAPDH expression as reference gene and presented as relative expression to control without DR agonist. HC *n* = 5, RA *n* = 4–6. F = Fenoldopam; R = Ropinirole; 6 = 10^−6^ M; 7 = 10^−7^ M; 8 = 10^−8^ M. Single dots represent mean values of duplicates for individual patients. * = *p* < 0.05 calculated by Wilcoxon matched pairs signed rank test compared to untreated control.

**Figure 9 cells-11-01609-f009:**
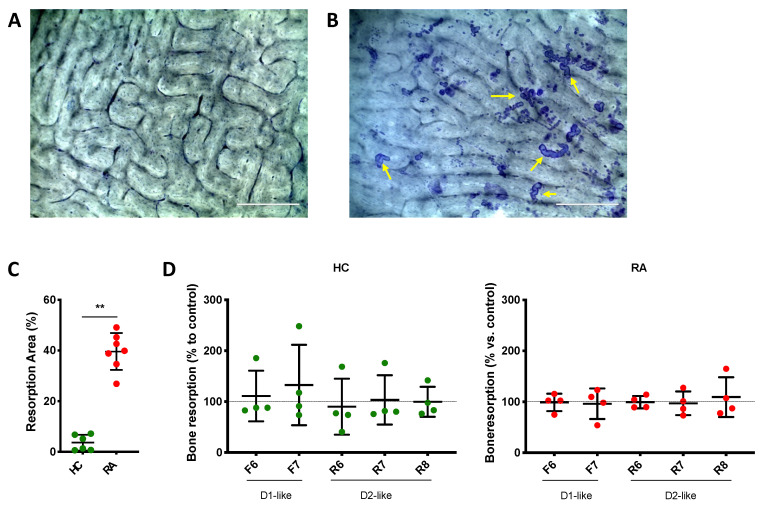
DR stimulation does not affect bone resorption. PBMCs were cultured on top of bone slices with OC differentiation medium for 14 days in the presence or not of DR agonists. Cells were then removed, and bone resorption pits were stained with toluidine blue. Percent of resorbed area compared to the total area was evaluated in three pictures for each condition. (**A**,**B**): representative pictures on toluidine-stained bone slices from a healthy donor. (**A**): empty bone, cultured without OC. (**B**): Bone slice culture with OC but without DR agonist. Yellow arrows indicate some of the resorption area present on the slice, as example. (**C**): percentage of bone resorption in cells from RA patients (red dots) and healthy controls (HC, green dots) cultured without DR agonists. (**D**): percentage of bone resorption in cells treated with DR agonists. Percentage of total resorbed area was normalized to control without DR agonist. HC *n* = 4–6, RA *n* = 4–7. F = Fenoldopam; R = Ropinirole; 6 = 10^−6^ M; 7 = 10^−7^ M; 8 = 10^−8^ M. Single dots represent mean value of individual patients. ** = *p* < 0.005 calculated with Mann Whitney test.

**Table 1 cells-11-01609-t001:** Subject characteristics.

	Bone		Blood	
	OA (*n* = 14)	RA (*n* = 14)	HC (*n* = 8)	RA (*n* = 9)
Female (%)	64	86	50	33
Age mean (years)	74	61	34	56
Age range (years)	52–85	26–87	22–59	40–67
Disease duration (mean, years)	N.A.	12	N.A.	11
Medication (% of total patients)				
Glucocorticoids	N.A.	71	N.A.	56
csDMARDs	N.A.	64	N.A.	67
bDMARDs	N.A.	43	N.A.	67

## Data Availability

The datasets used and/or analyzed during the current study are available from the corresponding author on reasonable request.

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
