# Peer review of "Modulation of Dopamine Receptors on Osteoblasts as a Possible Therapeutic Strategy for Inducing Bone Formation in Arthritis"

_cells, 2022, doi:10.3390/cells11101609_

Round 1

Reviewer 1 Report

In the part of introduction, the pathogenesis about RA and osteoporosis may be discussed.

RA related bony loss included marginal bone loss and systemic bone loss, and the author should evaluate and discuss.

The difference of marginal bone loss and systemic bone loss ? the author discussed the development of osteoporosis in RA, however, the marginal bone loss was important in the active destruction of bone in RA patients.  

The importance of systemic bone loss related osteoporosis in RA should be discussed.

The expression of D2-like DR in RA cells was associated with the disease activity of RA ? or the medication of biologics or DMARDs ? or RA related presentation ?

The author could not clear point the dopaminergic pathway as a therapeutic target  for inducing bone formation, and the conclusion in the abstract should be corrected.  

Author Response

In the part of introduction, the pathogenesis about RA and osteoporosis may be discussed.

RA related bony loss included marginal bone loss and systemic bone loss, and the author should evaluate and discuss. 

The difference of marginal bone loss and systemic bone loss ? the author discussed the development of osteoporosis in RA, however, the marginal bone loss was important in the active destruction of bone in RA patients.  

The importance of systemic bone loss related osteoporosis in RA should be discussed.

 Thank you for these precious suggestions. We added a paragraph in the introduction describing the different forms of bone loss in RA and their pathogenesis.

The expression of D2-like DR in RA cells was associated with the disease activity of RA ? or the medication of biologics or DMARDs ? or RA related presentation ?

This is a very interesting question. Unfortunately, no clinical data are available for the cohort we used, and 90% of the patients used for DR FACS analysis were under biologics. We can therefore not perform the suggested correlations.

The author could not clear point the dopaminergic pathway as a therapeutic target  for inducing bone formation, and the conclusion in the abstract should be corrected.  

Thank you for this comment. We corrected the sentence in order to avoid misunderstanding.

Reviewer 2 Report

The manuscript submitted by Schwendich et al described the involvement of dopamine and its receptors in the modulation of osteoblasts and osteoclasts in RA patients, providing an interesting contribution to the current knowledge of the role of dopaminergic signaling in human bone homeostasis. This study further helps to understand  the role of D2-like receptors in RA bone remodeling by showing experimentally that D2-like agonists are able to induce bone matrix formation and to affect OC differentiation, suggesting its potential as a novel therapeutic target. However, the following concerns with the manuscript need to be addressed:

  1. Although OA samples have classically been included in studies on the pathogenic mechanisms underlying RA, the variability of the results obtained in the present study evidences its questionable validity as a reference entity to compare with. The expression of the system of dopamine and its receptors should have been studied in osteoblasts from healthy subjects. In fact, in this same manuscript, the authors use cells from healthy controls to describe in a comparative manner the effect of dopamine receptor agonists in osteoclasts of RA patients. Why this difference in the experimental design between OBs and OCs?
  2. In the different experiments involving the use of D1-like and D2-like receptor agonists, a range of concentrations of these agonists have been used, including considerably high levels of both Fenoldopam and Ropinirole. However, apparently it has not been tested whether these concentrations influence the viability/proliferation of the cells studied. The authors should address this question, by providing evidence that neither Fenoldopam nor Ropinirole significantly affects the viability/proliferation of stimulated cells
  3. Results reported in Figures 5 and 6 describe differential effects of D2-like receptor agonists on primary osteoblast cultures and activated osteoblasts. However, this question is not discussed in detail. In this regard, it would be interesting to characterize the expression of dopamine receptors in both conditions: is there a variation in the expression levels of D2-like receptors? How do the authors explain these differential effects?
  4. This paper demonstrates that Ropinirole, a D2-like receptor agonist,stimulates osteoclastogenesis both indirectly (inducing the production of MIF and RANKL by OBs) and directly (increasing TRAP expression in OCs and the number of TRAP+ cells) in cells isolated from RA patients. These results are opposite to the inhibitory effect of D2-like receptor-mediated signaling on osteoclastogenesis described by Hanani et al. (reference 35) in healthy donor cells. Similarly, the results in Figures 7,8 and 9 show an absence of effect of dopaminergic signaling on OCs from healthy controls. How do the authors explain these differences?
  5. Since the presence of dopamine receptors is limiting in the mediation of their effects, it is essential to characterize the expression of the different receptor subtypes in the cells under study. Therefore, as suggested in a previous point, a detailed analysis of the expression of dopamine receptors in the distinct functional states of RA osteoblasts should have been performed. Similarly, it is crucial to characterize the presence of these receptor subtypes in both precursors and OC from RA patients. The availability of this information would greatly contribute to the interpretation of the results.

Some minor concerns are specified below:

  1. Figure 3B: The legend of the ordinate axis is not clear. does it refer to pg/10^6 cells?
  2. Figure 4: Section A of this figure does not provide any information that is not reflected in the other sections of the figure. These images should be included as supplementary material in the manuscript.
  3. Line 353: In this line it is stated “…the increase in mineralization after D2-like DR stimulation was detected in all RA patients, thus…”. However in Figure 4D, column R8 contains points below the control. Authors should specify that the inducing effect is observed at high concentrations of the agonist.
  4. Line 425 : “No significant effects of DR stimulation were observed in OA (fig 7C)” Results included in Figure 7 concern cells from RA patients and healthy controls, therefore, it seems that the authors have been mistaken and have spelled OA instead of HC.
  5. It would be interesting for the authors to discuss the fact that D1/D2 like receptor agonists reduce the expression of the transcription factor NFTAc1 in RA cells but increase that of TRAP which is transcriptionally regulated by NFATc1.
